# The “Yin and Yang” of Unfolded Protein Response in Cancer and Immunogenic Cell Death

**DOI:** 10.3390/cells11182899

**Published:** 2022-09-16

**Authors:** Nicole Rufo, Yihan Yang, Steven De Vleeschouwer, Patrizia Agostinis

**Affiliations:** 1Laboratory of Cell Death Research & Therapy, Department of Cellular and Molecular Medicine, KU Leuven, 3000 Leuven, Belgium; 2VIB Center for Cancer Biology Research, 3000 Leuven, Belgium; 3Research Group Experimental Neurosurgery and Neuroanatomy, Department of Neurosciences, KU Leuven, 3000 Leuven, Belgium; 4Department of Neurosurgery, University Hospitals Leuven, 3000 Leuven, Belgium; 5Leuven Brain Institute (LBI), KU Leuven, 3000 Leuven, Belgium

**Keywords:** ER stress, UPR, inflammation, cancer, immunogenic cell death, ICD

## Abstract

Physiological and pathological burdens that perturb endoplasmic reticulum homeostasis activate the unfolded protein response (UPR), a conserved cytosol-to-nucleus signaling pathway that aims to reinstate the vital biosynthetic and secretory capacity of the ER. Disrupted ER homeostasis, causing maladaptive UPR signaling, is an emerging trait of cancer cells. Maladaptive UPR sustains oncogene-driven reprogramming of proteostasis and metabolism and fosters proinflammatory pathways promoting tissue repair and protumorigenic immune responses. However, when cancer cells are exposed to conditions causing irreparable ER homeostasis, such as those elicited by anticancer therapies, the UPR switches from a survival to a cell death program. This lethal ER stress response can elicit immunogenic cell death (ICD), a form of cell death with proinflammatory traits favoring antitumor immune responses. How UPR-driven pathways transit from a protective to a killing modality with favorable immunogenic and proinflammatory output remains unresolved. Here, we discuss key aspects of the functional dichotomy of UPR in cancer cells and how this signal can be harnessed for therapeutic benefit in the context of ICD, especially from the aspect of inflammation aroused by the UPR.

## 1. The UPR and Its Main Branches

The unfolded protein response (UPR) was identified about 30 years ago [1] in yeast and subsequently, recognized as an evolutionary conserved signaling pathway in metazoan. The UPR is a signaling pathway evoked by the accumulation of unfolded or misfolded proteins in the ER lumen caused by perturbations of the multiple functions of the ER, namely, protein folding and secretion, lipid synthesis and Ca^2+^ storage. From an evolutionary point of view, the overarching aim of the UPR is to sense functional alterations within the lumen of the ER and engage a cytosol-to-nucleus signaling pathway that relieves the ER protein burden and restores ER homeostasis.

The UPR entails the activation of three entwined signaling branches, governed by three ER transmembrane effectors, PERK, IRE1α and ATF6 [2], which will be briefly discussed below. These ER sensors consist of three domains, namely, an ER luminal domain (responsible for sensing unfolded peptides), a single-pass transmembrane domain and a cytosolic domain, which, in the case of PERK and IRE1α, possesses catalytic activity. Upon perturbation of ER homeostasis, the glucose-regulated protein GRP78/BiP (or just BiP) [3], serving as the master key of inactivity of stress sensors, sets these sensors free to perform their downstream functions (Figure 1). Crystallization studies [4,5,6] show a high degree of similarity between the luminal domains of PERK and IRE1α either in yeast, mice or humans, suggesting common and conserved mechanisms of action. However, despite extensive research, the exact mechanism that senses alteration in the folding machinery and couples it to the activation of these ER stress sensors is still elusive. Two main models have been proposed: one proposes that the dissociation of BiP caused by the accumulation of client proteins would buffer BiP away from the luminal domains of PERK, IRE1 and ATF6, thus driving their activation (as depicted in Figure 1); alternatively, direct binding of unfolded proteins/peptides to the luminal domain of the ER stress sensors would promote their activation [7]. In fact, these two models may coexist and be further regulated by additional signals affecting the lipid composition of the ER membrane (as discussed in [8,9]).

PERK is rapidly activated following ER stress. In its active conformation, PERK was predominantly found as a dimer, but it can transiently form a tetramer with increased phosphorylation activity that could represent a key step in UPR induction [5]. Activated PERK phosphorylates eukaryotic translation initiator factor 2α (eIF2α) that leads to rapid and reversible attenuation of protein synthesis [10,11]. Despite the general shutdown of protein translation, phosphorylation of eIF2α triggers the translation of a subgroup of mRNAs containing short open reading frames in their 5ʹ untranslated regions, such as that of the activating transcription factor 4 (ATF4) [12]. In turn, ATF4 upregulates ER chaperones and foldases as well as proteins involved in redox processes and amino acid metabolism. Once ER stress is resolved, protein translation is resumed by dephosphorylation of eIF2α mediated by the activity of protein phosphatase 1 (PP1) that oligomerizes with a substrate-specific PPP1R15 regulatory subunit (that possesses two isoforms, GADD34 and CReP). Recent evidence supports the need for a third component in the complex, i.e., monomeric G-actin, to fully determine substrate specificity and increase affinity towards eIF2α [13,14]. PERK may also counteract oxidative stress by phosphorylating and consequently, triggering the dissociation of the nuclear factor erythroid 2-related factor (NRF2) from Kelch-like ECH-associated protein 1 (KEAP1) and the activation of genes harboring antioxidants response elements (ARE) in their promoter [15]. Of note, eIF2α can be phosphorylated independently of ER stress by three other protein kinases with high homology to PERK, namely, general control nonderepressible 2 (GCN2, activated by amino acid deprivation), protein kinase R (PKR, activated by double-stranded RNA) and the heme-regulated inhibitor kinase (HRI, activated by oxidative stress). These eIF2α regulatory pathways are referred to as an integrated stress response (ISR) that allows channeling of different metabolic stresses into a common hub [11].

IRE1 is the most conserved evolutionary branch of the UPR. There are two mammalian homologs of IRE1: IRE1α, which is expressed in all cell types, and IRE1β, whose expression is restricted to the mucosal epithelium, such as respiratory and gastrointestinal tracts. ER stress leads to IRE1 dimerization, triggering its serine/threonine-protein kinase activity and trans-autophosphorylation. In turn, this event activates the RNase domain of IRE1 by conformational rearrangements. Active IRE1 catalyzes the excision of a 26 nucleotide intron from the X-Box binding protein 1 (XBP1) mRNA, causing a frameshift that removes a premature stop. This unconventional splicing leads to the translation of the full-length XBP1 mRNA, which encodes a transcription factor that controls genes involved in protein folding and secretion, ERAD, lipid synthesis and redox homeostasis [16]. While XBP1 is the only known splicing target, IRE1 can also degrade a subset of mRNAs through a process known as regulated IRE1-dependent decay (RIDD). How RIDD activity is regulated is not fully understood. The degradation of mRNAs found in the proximity of the ER, often encoding proteins belonging to the secretory pathway, may serve as a mechanism to reduce mRNA abundance and ER protein folding. The kinetics of XBP1 splicing and RIDD are not coinciding, hinting at a differential regulation of the RNAse activity possibly due to a different oligomerization status [17].

Upon ER stress, ATF6, which exists in two isoforms (ATF6α and ATF6β), exposes the Golgi localization signal and moves to the Golgi where it is cleaved at two sites by the site 1 and site 2 proteases (S1P and S2P). The N-terminal cleaved fragment, called ATF6p50, then translocates to the nucleus where it forms active homodimers or dimerizes with other transcription factors such as nuclear transcription factor Y (NF-Y) as well as XBP1s, where it mainly induces the expression of genes of the ERAD pathway, lipid biosynthesis and XBP1 itself [18]. Together, XBP1 and ATF6p50 also increase ER and Golgi biogenesis to recover the secretory capacity of the ER [8,19].

The threshold of ER stress regulating the fine-tuning of these three partially overlapping pathways is fundamental for cell survival during ER stress. However, the signal integration mechanisms that govern the ultimate UPR output [2] are complex and not fully elucidated yet. Indeed, each ER stress sensor is further controlled by its interactome and post-translational mechanisms, which may impact the kinetics and amplitude of each arm of the UPR [8,20].

## 2. The Prosurvival Function of the UPR

Once activated, the multifactorial adaptive processes driven by these UPR signal transducers can be summarized as follows (Figure 2). First, the entrance of ER client proteins is decreased through the degradation of ER membrane-associated mRNAs by IRE1-RIDD as well as through the attenuation of protein translation by PERK-eIF2α. Second, the ER volume is enlarged by de novo lipid synthesis [19,21] and repopulated by newly synthesized ER chaperones and foldases to increase the folding capacity of the ER. Third, the turnover of misfolded protein is elevated by the increased transcription of ERAD-related proteins. Finally, possible causes of ER stress are buffered via upregulation of antioxidant or metabolic genes. In addition to strictly supporting and enhancing the ER folding machinery, the UPR sustains survival by promoting autophagic flux. Autophagy supports the degradation of misfolded proteins and protein aggregates as a noncanonical ERAD pathway. The PERK axis, through ATF4 and the C/EBP homologous protein (CHOP), can regulate autophagy by upregulating autophagic (ATG) genes [22], and the IRE1 intersects with the autophagy pathway through its direct downstream effector XBP1 or via the scaffolding function of the adaptor protein TNF receptor-associated factor 2 (TRAF2) [23].

UPR sensors are also closely linked to mitochondrial dynamics. ATF4 is a regulator of Parkin [24], which controls mitochondria clearance and dynamics, whereas ATF6 can act as a coactivator with the master regulator of mitochondria biogenesis peroxisome proliferator-activated receptor gamma coactivator 1α (PGC1α) [25]. Both PERK and IRE1 are moonlighting proteins, with UPR-independent functions, at the proteinaceous domain of close appositions between the membranes of the ER and mitochondria called ER mitochondria contact sites (ERMCs) [26]. In resting cells, the association of PERK to ERMCs regulates ER–mitochondria lipid transfer, Ca^2+^ signaling and mitochondria respiration (*Sassano* et al., *unpublished data*). Likewise, ERMC-associated IRE1 favors the transfer of Ca^2+^ from ER to mitochondria by physically interacting with the inositol triphosphate receptors (IP3Rs) [27], thereby supporting mitochondria bioenergetics and ATP production through the tricarboxylic acid cycle under steady state. PERK is implicated in the control Ca^2+^ dynamics, also through its interaction with the actin-binding protein filamin A. Under conditions of ER stress caused by ER-Ca^2+^ store depletion, a PERK–filamin A axis remodels the cytoskeleton and favors STIM1-mediated ER–plasma membrane contact sites, leading to store-operating Ca^2+^ entry (SOCE) [28]. Since SOCE is a mechanism to replenish luminal ER Ca^2+^ storage after stimuli causing its depletion, PERK may regulate the amplitude and spatiotemporal control of the Ca^2+^ signal between the ER and mitochondria with important implications for mitochondria homeostasis and survival or death decisions [29]. 

Hence, beyond their established role in coordinating the adaptive UPR, ER stress sensors modulate cell fate by additional noncanonical processes. The mechanistic underpinning regulating their recruitment at ERMCs, the distinct role they play under conditions eliciting ER stress and their interacting partners within these domains remain an area of intense research.

## 3. The UPR as a Mechanism of Cell Death 

The cell death module of the UPR (known also as terminal UPR) is elicited by the release of proapoptotic mitochondrial proteins such as cytochrome c and Smac/DIABLO into the cytosol, driving the activation of caspases [30]. The terminal UPR pathway is largely governed by CHOP, which can behave either as a transcriptional activator or as a repressor [31] (Figure 2). 

Although both ATF6 and XBP1s are able to bind to the promoter of CHOP, the PERK-ATF4 axis appears to be crucial for dictating CHOP upregulation [32]. This is in line with the finding that upon severe ER stress, the PERK branch is rapidly engaged and sustained through the apoptotic process, whereas the IRE1 and ATF6 signaling pathways are mostly attenuated towards the final apoptotic phase [16]. CHOP mainly acts on the BCL2 gene family by repressing the transcription of the antiapoptotic members (i.e., Bcl-2, Bcl-XL, Mcl-1 and Bcl-W) and upregulating that of the proapoptotic members (i.e., BAX, BAK, BID, BIM, BAD, NOXA and PUMA). This process eventually results in the multimerization of BAX and BAK on the outer membrane of the mitochondria, leading to its permeabilization. CHOP also upregulates GADD34, which oligomerizes with PP1 and dephosphorylates eIF2α, thereby resuming translation and further increasing ER stress [31]. Additionally, CHOP drives the transcription of death receptors 4 (DR4) and 5 (DR5), which is counterbalanced by IRE1-RIDD-mediated DR5 mRNA degradation [33]. In the case of unresolvable stress, the PERK-CHOP axis prevails, and DR5 accumulation leads to ligand (i.e., tumor necrosis factor (TNF)-related apoptosis-inducing ligand (TRAIL))-independent multimerization that, in turn, accelerates the formation of the death-inducing signaling complex (DISC) and activates caspase-8. In a recent study, glucose deprivation has been shown to drive apoptotic (and partially necrotic) cell death mediated by ATF4-dependent but CHOP-independent upregulation of DR5 [34]. The centrality of DR5 in caspase-8 activation during ER stress-induced apoptosis is still debated as there are opposing views that claim it as crucial [33,35] or dispensable [36], despite using the same ER stressor in the same cell lines. However, it is possible that when DR5 is artificially removed, DR4 or other death receptors could have compensatory roles. DR5 has also recently been shown to directly sense misfolded proteins and promote apoptosis [37]. 

ATF4 also possesses CHOP-independent proapoptotic functions by driving the transcription of ubiquitin ligases that promote the degradation of the antiapoptotic XIAP [38]. IRE1 can also independently contribute to apoptosis by further increasing its RIDD program leading to the degradation of more mRNA encoding for ER-localized enzymes [39] or miRNAs that normally repress proapoptotic proteins such as thioredoxin-interacting protein (TXNIP) [40]. Moreover, IRE1 transphosphorylation has been discovered to possess scaffolding functions independently from the RNAse activity that leads to the recruitment of adaptor protein TNF receptor-associated factor 2 (TRAF2) [41]. TRAF2, in turn, activates JNK, a member of the mitogen-activated protein kinase (MAPK) superfamily, that activates several proapoptotic proteins such as p53, BAD and BIM by phosphorylation [42].

In terms of ER stress, cell fate regulation by PERK and IRE1 is also decided through their noncanonical and UPR independent role as components of the ERMCs. Under conditions of ER oxidative stress, PERK promotes the rapid transfer of reactive oxygen species (ROS) from the ER to the mitochondria, facilitating rapid cardiolipin oxidation and cytochrome c release [43]. Under mild ER stress, ubiquitylation of IRE1 by the ubiquitin ligase membrane-associated ring-CH-type finger 5 (MARCH5) at the ERMCs inhibits IRE1 oligomerization and its prodeath RIDD activity [44]. When ER stress persists, IRE1 ubiquitylation is reduced, and cell death ensues [44]. This suggests that stress-mediated post-translational mechanisms operate at the ERMCS to control IRE1 function and the switch between its prosurvival and proapoptotic roles. Recent interactome studies indicate that protein interactions can have a profound effect on IRE1 and PERK [45]. For example, Bax inhibitor-1 (BI-1) and fortilin can bind to phosphorylated IRE1 and decrease its signal output [46,47]. In contrast, IRE1 signaling is sustained by direct interaction with members of the B-cell lymphoma 2 (BCL-2) family to promote apoptosis [48]. PERK interacts with several proteins involved in cytoskeleton remodeling, such as filamin A and lipid binding and transfers proteins located at the membrane contact sites among others [49], although the functional roles of the interacting partners of PERK still need to be fully unraveled.

While apoptosis is the main outcome of the lethal action of the UPR, recent studies suggest that nonapoptotic cell death pathways can also be induced by ER stress. For example, thapsigargin (a well-known ER stress inducer that depletes Ca^2+^ from the ER) in a murine fibrosarcoma cell line can drive regulated necrotic (necroptotic) cell death mediated by TNFR1 in a ligand-independent fashion through RIPK1, RIPK3 and MLKL. Interestingly, removal of these proteins would not prevent cell death but redirect it through the apoptotic machinery [50]. Recently, activation of the UPR and in particular, of the PERK arm [51,52], has been shown to participate to the regulation of ferroptosis, an iron-dependent and lipid peroxide-driven necrotic cell death [53]. However, the mechanistic underpinning and the primary contribution of ER stress in ferroptosis remains to be validated.

While persuasive evidence links unresolved ER stress to mitochondria apoptosis, it remains unclear whether the UPR is activated as a secondary response to other forms of cell death. 

## 4. Proinflammatory Pathways Driven by the UPR 

Following ER stress, all the three UPR branches can independently contribute to the activation of NF-κB, the master regulator of proinflammatory responses (Figure 3).

The PERK-eIF2α-mediated attenuation of translation leads to an imbalance between NF-κB protein levels and the short-lived IκBα favoring the presence of inhibitor-free NF-κB that is then able to translocate to the nucleus [54]. IRE1 forms a complex with IKK through the recruitment of the adaptor protein TRAF2, leading to IκBα degradation and NF-κB activation. IRE1 can also trigger the activation of the three members of the MAPK family. Indeed, TRAF2 acts as a scaffold to recruit apoptosis signal-regulating kinase 1 (ASK1), a MAP3K that, in turn, activates the p38 MAPK and JNK pathway [55], leading to phosphorylation of the activator protein 1 (AP1). AP1 is a dimer of proteins belonging to different families such as c-Fos, c-Jun, ATF and JDP that combine to regulate the transcription of proinflammatory genes [56]. IRE1 is also partially involved in the activation of ERK signaling upon ER stress by binding to the adaptor protein Nck [57]. Recently, it has also been proposed that the IRE1-TRAF2 complex could lead to the activation of NF-κB through an additional pathway involving the cytosolic peptidoglycan receptors nucleotide-binding oligomerization domain containing 1 (NOD1) and NOD2 [58]. IRE1 has also been reported to indirectly activate the NF-κB pathway by the regulation of glycogen synthase kinase 3 [59]. ATF6 can, instead, support NF-κB activation through the mTOR pathway and AKT dephosphorylation [60]. 

In addition to the activation of canonical proinflammatory pathways, signaling components of the UPR signaling machinery can directly drive transcription of proinflammatory genes. For example, XBP1s have been found associated with the promoter region of the gene encoding for TNFA, IL6 and IFNB1 [61,62], while ATF4 and CHOP have been found to be associated with the promoter region of IL6 and IL23A, respectively [63,64]. ATF6 has also been reported to dimerize with cyclic AMP-responsive element-binding protein 3-like protein 3 (CREBH) in the liver to direct the transcription of genes involved in the acute phase response possibly igniting a systemic proinflammatory response [65]. 

In addition to the canonical UPR stress sensors, DR4 and DR5 have also been recently endowed with a proinflammatory feature [66]. Indeed, cell treatment with different ER stressors (such as taxanes or the classical ER stress insults Brefeldin A and thapsigargin) led to ATF4/CHOP-dependent upregulation of DR5 that caused the activation of the NF-κB signaling cascade through a pathway involving the FADD/caspase-8/RIPK1 (FADDosome) complex [66]. As DR5 activation was ligand-independent, it was suggested that activation of the proinflammatory pathway is triggered by the elevation of DR5 over a threshold level. Of note, the assembly of the FADDosome complex was recently found to occur at the ER–Golgi intermediate compartment, and activation could be driven by the direct binding of DR5 to unfolded proteins [37].

## 5. Autocrine Role of the UPR in Tumor Cells

Tumor cells need to survive in a hostile tumor microenvironment (TME), and chronic activation of the prosurvival axis of the UPR has emerged as a crucial hallmark of cancer. The major stressors persistent in a tumor microenvironment include (but are not limited to) severe hypoxia, nutrient deprivation and acidosis, which eventually induce (chronic) ER stress in cancer cells. Moreover, in addition to these extracellular stressors, the ER is also challenged by intracellular factors such as oncogenic activation that requires UPR to escape oncogene-induced apoptosis and to support the high demand for protein synthesis. Altogether, these stressful conditions exacerbate the ER protein folding machinery inducing ER stress/UPR [67]. Chronic ER stress eventually exerts a selection pressure on cancer cells. This will favor the predominant persistence of cancer cells that not only can cope with the presence of chronic ER stress but also utilize its maladaptive function to promote growth, while at the same time actively blunting signaling pathways associated with UPR-mediated cell death. Several mutations in oncogenes and onco-suppressor genes [68], as well as in the genes involved in the UPR machinery itself [69,70], have been demonstrated to prevent UPR-induced apoptosis. It has been reported that mutations of IRE1 can convert it into an oncogene [71]. Similarly, high levels of UPR components, such as BiP, PERK and XBP1 [72,73,74], are associated with poor prognosis in various cancer patients. In line with this, preclinical findings have demonstrated that experimental ablation of the key proteins of the UPR (PERK [75,76], IRE1 [77,78] and XBP1 [79]) results in impaired tumor growth. The protumorigenic ability of the UPR is exerted at different levels involving cancer cell autonomous and nonautonomous mechanisms. 

BiP upregulation, downstream of both XBP1s and ATF6 signaling pathways, tends to counteract apoptosis by sequestering either caspase-7 [80] or the proapoptotic Bcl-2-interacting killer/BIK [81]. The PERK/ATF4 axis exerts a prosurvival function by promoting cytoprotective autophagy [82], inhibition of genes implicated in senescence [83] and upregulation of microRNA (miR) miR211 leading to downregulation of its target CHOP [84]. Increased ROS might lead to DNA instability and block in cellular proliferation, which is hindered by the antioxidant PERK-NRF2 branch [85]. Autophagy and antioxidant activity induced by PERK also support cancer cells in overcoming cell death (i.e., anoikis) elicited by cell detachment from the extracellular matrix, which is a necessary step for the formation of metastasis [86]. Formation of metastases is also favored by PERK-mediated upregulation of LAMP3 that supports migration and invasion [87], and levels of ATF4-regulated genes correlate with the epithelial-to-mesenchymal transition (EMT) signature in different tumor types [88]. 

The IRE1-XBP1 axis promotes tumor proliferation by upregulating cyclin A1 [89]. Tumor quiescence/dormancy is key for therapy resistance and tumor reoccurrence. ATF6 is found constitutively active in quiescent tumor cells where it supports dormancy by regulating the mTOR pathway [90]. PERK, instead, promotes dormancy by inhibiting cyclin D1 translation as a result of the global attenuation of translation, blocking the cell cycle in the G1 phase [10]. All three branches (through ATF4, XBP1s and ATF6) can directly bind to the promoter of VEGFA and other proangiogenic genes to alleviate tumor hypoxia [76,91]. In addition, stimulation of the UPR in tumor cells can mediate immunosuppression by inducing ER stress in neighboring immune cells, in particular, myeloid cells, through a process named “transmissible ER stress” [92]. This causes upregulation of pro-tumorigenic cytokines such as IL6, IL23 and TNF-α and secretion of the protumorigenic chemokines CCL3 and CCL4 as well as upregulation of Arginase 1 that suppresses T cell function.

## 6. The UPR in Immunogenic Cell Death 

Over the last 15 years, molecular and preclinical evidence from several laboratories challenged the dogmatic view that considered apoptosis strictly and inevitably as a tolerogenic form of cell death. A growing list of diverse anticancer therapies, including but not limited to, anthracyclines, radiotherapy, bortezomib, oncolytic viruses, photodynamic therapy, extracorporeal photochemotherapy and certain types of targeted therapies [93,94,95,96,97], can elicit a peculiar form of apoptosis, dubbed immunogenic cell death (ICD), which favors antigen-specific immune responses driving antitumor immunity (reviewed in [98,99]). Clearly other forms of regulated cell death, with robust inflammatory properties such as necroptosis and pyroptosis, are immunogenic [100,101]. However, the role of the UPR has not been firmly established in the latter contexts. Here, we will briefly introduce the main hallmarks of ICD driven by immunogenic apoptosis (reviewed in [102]).

ICD is hallmarked by the spatiotemporal surface relocation or release of danger molecules or damage-associated molecular patterns (DAMPs), which precedes or is concomitant to the cell death process. DAMPs are endogenous molecules with housekeeping functions in unstressed cells, which once exposed to the extracellular environment in response to cellular stress or injury, act as danger signals that are sensed by the immune system. By binding their cognate receptors on innate immune cells (e.g., professional antigen-presenting cells such as DCs), DAMPs favor the priming of the adaptive immune system and subsequent evocation of tumor antigen-specific CD8 T cell-mediated immune responses leading to the elimination of the residual cancer cells and the establishment of immunological memory (Figure 4). Scrutiny of the mechanistic underpinnings of ICD and several in vivo studies have unraveled that the main ICD-associated DAMPs include calreticulin (CRT), ATP and high mobility group box 1 (HMGB1).

The ER resident chaperone CRT, which is involved in protein folding, quality control and calcium homeostasis, is rapidly trafficked and externalized on the outer side of the plasma membrane (ecto-CRT) during the preapoptotic or early apoptotic phase. Ecto-CRT binds to CD91 on DCs and functions as a potent “eat-me” signal, facilitating the engulfing of dying cancer cells. A recent study highlighted how the adjuvant role of CRT is strictly associated with a limited spatiotemporal window [102]. Indeed, CRT mutations that lead to the loss of the KDEL ER retention peptide have been reported to cause continuous and uncontrolled extracellular CRT release. In turn, this causes decreased phagocytosis of cancer cells by antigen-presenting cells (APC) possibly due to the saturation of the scavenger receptors and decreased anticancer efficacy in response to ICD inducers [102]. Secretion of ATP, the energy reservoir of the cell, occurs during the preapoptotic or early/late apoptotic phase of cell death and acts as a potent short-range “find-me” signal by binding the ionotropic P_2_RY_2_ purinergic receptors on DCs and monocytes. Secreted ATP can also bind the P_2_RX_7_ receptors on DCs and induce NLRP3/ASC/caspase-1 inflammasome-mediated IL1β release [103]. The release of the calcium- and phospholipid-binding protein annexin A1 (ANXA1) favors the homing dead cell (DC) synapse by binding to formyl peptide receptor 1 (FRP1) [102]. HMGB1 is passively released during the late apoptotic phase, and it acts as a “find-me” signal by binding the receptor for advanced glycosylation end products (RAGE) receptor on DCs. Moreover, it can also bind to TLR4 and facilitates antigen processing in DCs as well as activating the production of proinflammatory cytokines. The role of these DAMPs as critical molecular effectors of the dialogue between stressed/dying cancer cells and the immune system has been validated in different preclinical and clinical studies [96,104]. Understanding the mechanisms linking cellular stress and death pathways evoked by anticancer therapies to the release of these danger signals could help with designing therapeutic strategies to accentuate or promote their release using poorly immunogenic treatments.

The ER is physiologically and evolutionary programmed to communicate with the extracellular space in a relatively quick manner. Therefore, it is perhaps not surprising that UPR sensors orchestrate the trafficking of key DAMPs, such as CRT and ATP, on the surface of the stressed cancer cells succumbing through ICD as a mechanism to alert the immune system. Surface exposure of CRT in cancer cells responding to anthracycline mitoxantrone, a prototype ICD inducer, involves the concomitant activation of three modules: (i) an ER stress–ROS signaling mediated by the activation of the PERK-eIF2α axis; (ii) a cell death axis involving the preapoptotic cleavage of BAP31 by caspase-8 and regulated by BAX/BAK and Ca^2+^; and (iii) a SNARE-dependent, ER-to-Golgi anterograde secretory pathway [105]. ATP is secreted through a pannexin 1 and lysosome-dependent mechanism [106]. Instead, the stress pathway engaged upon PDT with hypericin, another well-studied paradigm of ICD, relies on PERK (but not eIF2α), BAX/BAK, Ca^2+^ and actin-mediated secretory pathways for the concomitant exposure of CRT and ATP [107]. The reason behind the stronger reliance of the danger signaling on PERK rather than other UPR sensors remains unknown. The UPR-independent role of PERK in mediating the juxtaposition of the ER membrane with the plasma membrane by interacting with filamin A [28] could possibly favor the externalization of DAMPs through maintaining Ca^2+^ signaling and the rearrangement of the cytoskeleton. However, the centrality of PERK has been recently been challenged by other studies describing that ROS-mediated ER and/or Golgi damage drives ICD involving other integrated stress responses of eIF2α kinases, GCN2, PKR or HRI, rather than PERK [108,109] and the activation of IRE1 and the ATF6 axis [109].

Thus, while the UPR is mechanistically linked to the exposure of danger signals from stressed/dying cancer cells, the choice of the trafficking mechanism harnessed by the ICD inducer is dictated by the intracellular damage and the ensuing stress pathways evoked.

## 7. Inflammation and ICD

Recent reports indicate that the immunological readouts of ICD cannot be exclusively explained through the release of the abovementioned DAMPs. Optimal immunogenicity may require the adjuvanticity provided by other factors synthesized *de novo* during the process of cell death. Indeed, ICD inducers, as triggers of ER stress, can activate inflammatory pathways which are either directly governed by the UPR or activated secondary to ER stress-mediated cell death. The ICD-associated proinflammatory output can eventually modify the extracellular milieu and immune cell recruitment and activation. Anthracyclines can activate the IFN response in the stressed/dying cancer cells, leading to the production and secretion of Type I interferons and CXCL10 [110]. Anthracyclines also induce the release of CCL2, necessary for the recruitment of antigen-presenting cells [111]. Production and secretion of CXCL8 has been shown as necessary for the surface relocation of CRT upon treatment with mitoxantrone [112], indicating a feedforward loop. Melphalan, a regional chemotherapy for locally recurrent metastatic melanoma, triggers the release of CXCL8, IL6, IL1β and CCL2 by murine melanoma cells. Together with the surface exposure of the chaperone and danger signal HSP90, these proinflammatory factors are able to elicit a partial activation and maturation of DCs and achieve protection in prophylactic vaccination settings despite the absence of ecto-CRT and ATP [113]. The concomitant release of the neutrophil chemoattractants CXCL1, CCL2 and CXCL10 by murine cancer cells is a shared hallmark of the stress program elicited by photodynamic therapy (PDT) and mitoxantrone but not accidental necrosis or tolerogenic apoptosis [114]. Functionally, the corelease of chemokines increases in vivo neutrophil migration at the site of vaccination and stimulates the neutrophil-mediated killing of cancer cells in vitro [114]. Moreover, beyond the stress-induced preapoptotic release of ATP, ICD has been recently associated with the late apoptotic release of other nucleic acids. These include dsDNA that binds to TLR7/8/9 on innate immune cells (such as neutrophils) regulating their activation and anticancer activity [114] and dsRNA that, by binding to TLR3 on other cancer cells, induces the production and release of type I interferon and in turn, CXCL10 [110]. 

These studies argue that sterile damage leading to cancer cell death by ICD can be sensed and decoded by the immune system in a manner similar to the immune sensing of pathogen-infected cells. Indeed, there are several analogies in common between ICD and pathogen infection. First, pathogens lead to the activation of the UPR and ISR as well as autophagy. These stress pathways, as discussed above, are determinants of the molecular machinery employed for externalization of key DAMPs acting as “eat me” (CRT) and “find me” (ATP, various nucleotides) signals, which allow fast immune recognition and phagocytosis of stressed/dying cells. In addition, DAMPs and pathogen-associated molecular patterns (PAMPs) share the same pattern recognition receptors (PRRs) [99]. In addition, type-I interferons have antitumorigenic properties evolved as a defense towards viral infections. Moreover, the chemokine signature selectively identified upon induction of ICD is reminiscent of the chemokine pattern elicited by bacterial or viral infections [114]. Malignant cells and pathogens have developed common strategies to subvert recognition by the immune system (reviewed in [109]). 

In a recent RNA-Seq profile study, the inflammatory output associated with robust ICD inducers (mitoxantrone, PDT) in human melanoma cells was found to require concomitant activation of NF-κB and AP-1, which led to the expression of several shared pro-inflammatory chemokines. This early stress pathway dissociated from ER stress-induced cell death and the UPR and was coordinated by HSP60 [115]. Strikingly, the IRE1 kinase inhibitor KIRA6 overruled the NF-κB/AP-1-mediated chemokines expression and release by targeting HSP60. The reduced in vivo vaccination potential of mitoxantrone-treated murine CT26 cells by KIRA6 suggests the relevance of NF-κB/AP-1-mediated inflammation for the efficacy of the anticancer vaccine, at least in this model of prophylactic vaccination [115]. This assumption is also supported by studies showing that the immunogenicity of necroptosis is mainly driven by the inflammatory program mediated by the RIPK1-NF-κB axis [116].

However, the magnitude, composition, temporal and spatial redistribution of cytokines and chemokines secreted by cancer cells in response to cellular damage regulate recruitment, activation of innate and adaptive immune effectors and their crosstalk [117]. Dendritic cells are recruited by CCL20 and CXCL12, whereas increased CXCL9 and CXCL10 secretion is associated with enhanced migration of CD8^+^ cytotoxic lymphocytes and NK cells that possess the cognate receptor CXCR3. CXCL12 and CCL2 recruit T_H_17 cells (by binding to CXCR4 and CCR6, respectively) that exert potent antitumor activities by recruiting CD8^+^ cytotoxic cells and DCs. On the other hand, regulatory and tumor-supportive T_reg_ lymphocytes migrate according to the gradient of CCL22 and CCL28 [118]. Overall, a bad prognosis is also associated with increased CCL2- and CCL5-dependent macrophage recruitment, as well as CXCL1/CXCL2/CXCL3/CXCL8-mediated chemotaxis of granulocytic MDSCs. Of note, ICD inducers can have a differential and direct impact on immune cells as well [119]. 

Altogether, these studies suggest that the proinflammatory output associated with ICD, whether directly emanating from the UPR or independent of it, may favor, in conjunction with DAMPs, a proficient dialogue between dying cancer cells and the immune system and accentuate the immunogenicity of ICD. However, the ultimate composition and balance between immunostimulating and immunosuppressive inflammatory mediators, their spatiotemporal release and local effectors will critically determine the local and peripheral immune responses against the tumor.

## 8. Therapeutic Outlook and Conclusions

Maladaptive UPR and loss or proteostasis have emerged as crucial hallmarks of cancer cells and the tumor microenvironment. For instance, a higher level of spliced XBP1 in myeloma patients indicates poor prognosis [120]. In glioblastoma where IRE1 somatic mutations have been linked to shorter patient survival, the IRE1 signaling dictates two distinct tumor phenotypes, with XBP1s driving the protumorigenic program, while RIDD activity attenuates it [121]. Likewise, constitutive activation of the PERK pathway has been linked to carcinogenesis and metastasis in different cancer types [122,123] (reviewed in: [124]). However, depending on the gene dose, PERK can function as either a tumor suppressor (when haploinsufficient) or a proadaptive tumor promoter [125]. While these data are consistent with the view that activation of the UPR in cancer supports malignancy, they also highlight the hurdles in unequivocally defining the role different UPR signaling branches play in carcinogenesis and thus their prognostic value as biomarkers. 

However, the protumorigenic function of the chronic activation of maladaptive UPR in cancer has therefore sparked interest in the development of small molecule inhibitors targeting components of the UPR machinery to impair cancer progression [126]. Screening studies have identified compounds that directly or indirectly target the UPR, which are currently used at the preclinical or clinical stage [127,128]. Small molecules have been developed that directly target the IRE1 RNAse domain (such as STF-083010 and 4μ8C) or compete with ATP binding, thus inhibiting the kinase autophosphorylation that drives dimerization and indirectly, the downstream of RNAse activity (as is the case for kinase-inhibiting RNase attenuators, KIRAs) [70]. Instead, the PERK branch can be targeted by inhibiting the kinase domain of PERK itself (with small molecules such as GSK2606414 [129] or the optimized GSK2656157 [130]) as well as by preventing the translation of ATF4 (with ISRIB) [131]. In vivo application of these UPR inhibitors has been found to impair tumor growth in several murine models [128,130,132]. However, it should be noted that the three UPR pathways are highly interconnected and therefore, the inhibition of one branch can be promptly compensated by the heightened activity of another one [133,134]. Alternative approaches for targeting tumors addicted to the UPR (e.g., myeloma cells) rely on inhibiting the proteasome or ERAD pathway to facilitate ER-associated proteotoxicity. For example, bortezomib, a selective inhibitor of the β5 subunit of the proteasome (PSMB5), has exhibited significant success in the treatment of patients with multiple myeloma and diffuse large B-cell lymphoma [135], probably because of the high secretory activity of these cells. Inhibitors of specific chaperones (such as BiP, heat shock protein 90/HSP90 and protein disulfide isomerase/PDI) have also been evaluated since “chaperone addiction” has been described as playing an important role in tumor transformation [126]. However, recent works reported that some of these inhibitors display UPR-independent activities and/or targets. For example, 4μ8C impacts insulin secretion [136] and possesses antioxidant properties [137], while KIRA6 and GSK2606414 share c-KIT as off-target. Moreover, the IRE1 kinase inhibitor KIRA6 targets various members of the heat shock family of proteins including, as mentioned above, HSP60 [115,138], thus raising the urgent need for a more careful characterization of the signaling mechanisms targeted by these chemical inhibitors of the UPR. Beyond this, there are several complexities associated with the targeting of the UPR in a tumor. As discussed, the UPR can display paradoxical effects on tumor progression depending on the context. In addition, caution should be exercised when targeting the UPR with inhibitors not specific for tumoral targets since other stromal cells rely on UPR components for their normal development and function, e.g., immune cells. For example, dendritic cells (DCs) are particularly dependent on UPR pathways. A subset of DC (CD8α+) critically depends on IRE1/XBP1s to elicit successful cross-presentation [139], whereas mucosal type 1 conventional DCs (cDC1s) depend on XBP1 for their survival in a tissue-specific fashion [140]. In another report, sustained XBP1 activation was shown to impair antigen presentation by tumor-associated dendritic cells [141]. Hence, the inhibition of ER stress pathways may variably interfere with tumor recognition and eradication by the immune system. 

While all these studies reveal the complexity of targeting the maladaptive, protumorigenic role of chronic UPR for cancer therapy, they also evoke the possibility to devise tactics that force the lethal arm of ER stress in cancer cells. As discussed in this review, the induction of terminal UPR by selective anticancer strategies kills cancer cells and favors the establishment of antitumor immunity responses. Thus, exacerbating, rather than inhibiting, ER stress seems to be a desired therapeutic approach. In particular, the design of smart therapies inducing ER stress-mediated cancer cell death, while sparing, if not even enhancing, antitumor immunity, by favoring the resetting of the immunosuppressive TME, seem particularly appealing. However, to fully explore the potential of ICD-based therapy, gaining a more precise molecular understanding of how therapy-induced UPR differs from the maladaptive UPR stimulated in cancer cells and how the immune system deciphers these signals, is a pressing need. In addition, recent studies highlight the necessity for careful identification of the secretome of cancer cells succumbing to ICD, the role of the UPR and its impact on the local and systemic immune responses. Finally, key ER stress sensors have additional UPR-unrelated roles as effectors of membrane contact sites, but research unravelling their noncanonical signaling mechanisms is still in its infancy. These are important future challenges which will require the integration of innovative technologies, including spatial omics and intravital imaging, in order to accelerate the translation of ICD therapy to the clinic.

## Figures and Tables

**Figure 1 cells-11-02899-f001:**
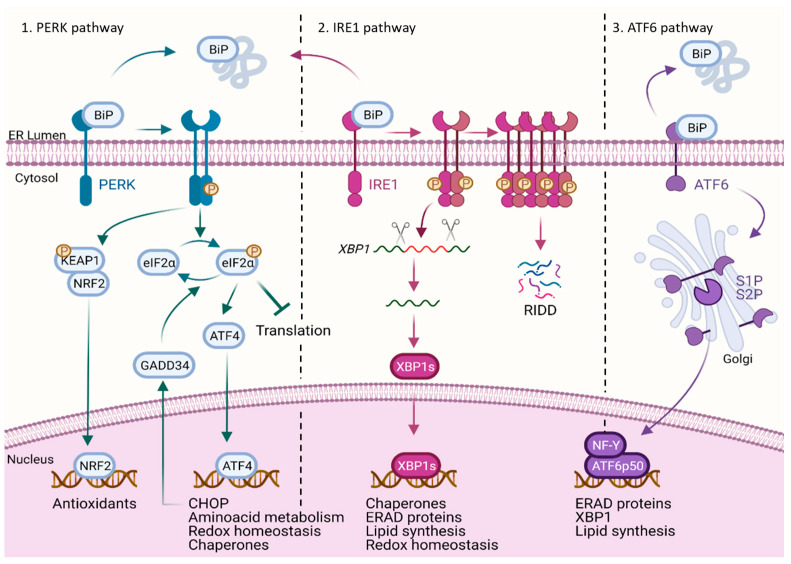
The three branches of unfolded protein response. In response to intra/extracellular stressors, protein folding capacity of ER is disturbed. BiP binds to the misfolded proteins in the lumen of the ER and sets the three UPR sensors PERK, IRE1α, and ATF6 free in the meantime. PERK dimerizes, autophosphorylates and then phosphorylates eIF2α. Protein synthesis will then be transiently inhibited in order to alleviate ER protein burden, except for specific mRNAs, such as ATF4. PERK may also dissociate KEAP1 from NRF2. IRE1 dimerizes and trans-autophosphorylates, activating its RNase domain to splice XBP1 mRNA into the more stable version, known as XBP1 mRNA, which encodes a potent transcription factor. IRE1 can degrade a subset of mRNAs through RIDD as well. ATF6 translocates to the Golgi where it is cleaved, and then its p50 cytoplasmic fragment heterodimerizes with NF-Y. Consequently, genes involved in ERAD, redox homeostasis and coding various chaperones are transcribed, first attempting to reverse the trend of protein folding turbulence and conserve the cells from dying.

**Figure 2 cells-11-02899-f002:**
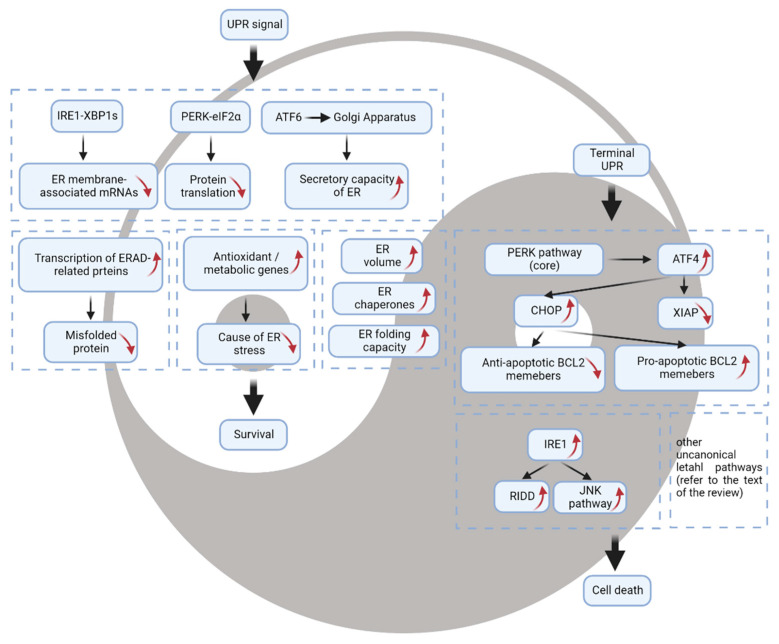
The “Yin and Yang” of the unfolded protein response. When cells experience loss of ER homeostasis, UPR will be activated, as a first attempt to attenuate protein load in the ER lumen and restore proteostasis. However, if the UPR fails to restore ER folding capacity, the UPR will enter the terminal phase to induce cell death. Up-arrow in red indicates up-regulation; Down-arrow in red indicates down-regulation.

**Figure 3 cells-11-02899-f003:**
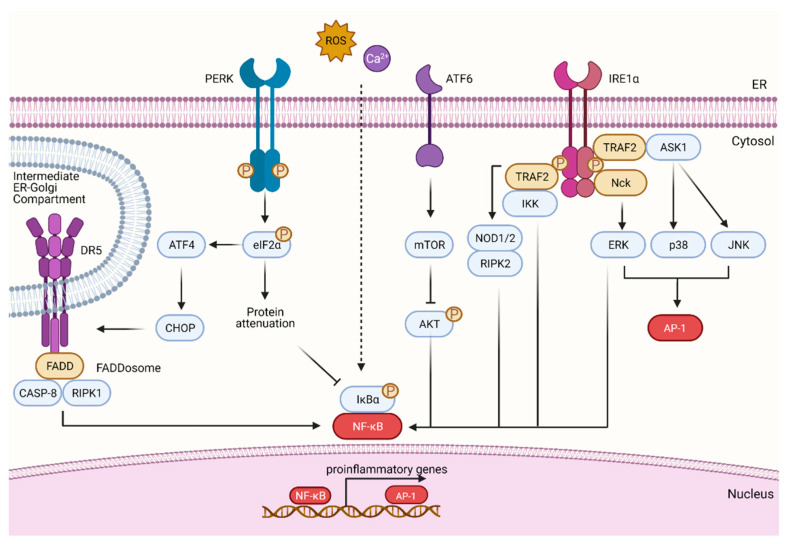
Proinflammatory pathways activated by ER stress. In the case of ER stress, Ca^2+^ release and ROS or UPR sensors may engage proinflammatory pathways. Indeed, through translation attenuation, PERK promotes NF-κB activation by reducing protein translation of its inhibitor IκBα. PERK/ATF4/CHOP axis is also responsible for DR5 upregulation that has been recently linked to the activation of NF-κB through the FADDosome. IRE1 can exert a scaffolding function leading to the assembly of a protein platform called UPRosome that comprises different adaptor and regulatory proteins, such as TRAF2, involved in proinflammatory processes. TRAF2 can activate NF-κB through NOD1, NOD2 and RIPK2 or by the recruitment and activation of IKK. TRAF2 can also recruit ASK1 leading to the activation of JNK and p38 and the downstream AP-1. IRE1α can also recruit Nck that activates ERK and in turn, NF-κB and/or AP1. ATF6 may also control NF-κB through mTOR-mediated AKT dephosphorylation.

**Figure 4 cells-11-02899-f004:**
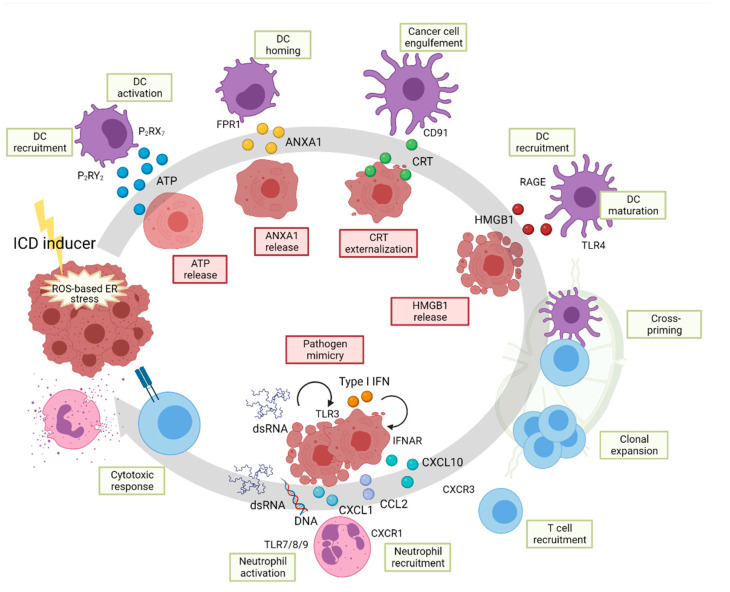
Immunogenic cell death (ICD). Upon treatment with ICD inducers, cancer cells experience ER stress caused by ER-associated ROS production that favors DC recruitment and activation through secretion of ATP (by binding to purinergic receptors P2RY2 and P2RX7, respectively) and their homing with released annexin A1 (ANXA1) binding to FPR1. Surface exposure of CRT stimulates the phagocytosis of dying cancer cells by engaging the CD91 receptor on DCs. In the late apoptotic phase, HMGB1 release recruits further DCs by binding RAGE receptor and induces their maturation through TLR4 signaling. DCs then migrate to the lymph node where they cross-prime and favor the clonal expansion of T cells. Cancer cells dying of ICD can mimic a pathogen infection response by releasing RNA that autocrinally induces type I IFN production and CXCL10 secretion (favoring the recruitment of tumor-specific T cells through CXCR3) and induce neutrophil recruitment and activation by releasing the chemokines CXCL1 and CCL2 as well as DNA and RNA. T cells and neutrophils then mediate the killing of live residual tumor cells.

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
