# Peer review of "The “Yin and Yang” of Unfolded Protein Response in Cancer and Immunogenic Cell Death"

_cells, 2022, doi:10.3390/cells11182899_

Round 1
Reviewer 1 Report
This appears a comprehensive and interesting review. I have listed a few smal language changes below. The main change I would suggest is to label the 3 branches discussed moe clearly in Fig 1 and maintain this numbering elsewhere throughoutthe text.
Line 30: use the term ‘branches’ rather than arms, to be consistent with the rest of the text
Line 40: remove ‘the’ before PERK, IRE1…
Figure 1: label the 3 branches more clearly. Would be best to use numbering 1-3 and maintain the same numbering throughout the manuscript.
Line 464: change ‘binds on’ to ‘binds to’
Author Response
Thank you for the detailed comments. We’ve revised the manuscript accordingly.
Reviewer 2 Report
1. Authors reviewed in article that chronic activation of the UPR signaling axis had become a hallmark of cancer. What was the practical significance for clinical screening and treatment of tumor diseases of this detection. Whether the activation of UPR signaling pathway could be used as an indicator to indicate the occurrence and progression of disease, or whether there were reported molecular targets in the signaling pathways involved in UPR that could be used for clinical treatment.
2. The structure of the article was not clear enough, it was better if the content in a logical structure could be combined with a summative title.
3. There was a lack of necessary foreshadowing and cohesion from one content to the next, and it was better to have a corresponding statement in the article. The same problem also existed at the end of each section, there was a lack of generalization statement of each section.
Author Response
Thank you for the helpful comments. We have added few sentences within the Therapeutic outlook and conclusions, providing some examples of UPR markers predicting the prognosis of cancer patient based on a published clinical trial, and others studies.
Regarding the comment of the Reviewer on the lack of generalization statements, we submit that while several sections do end with a conclusive statement, in our concluding remarks we do discuss several outstanding questions and challenges that remain still unanswered in the field of UPR and cancer and how to harness UPR pathway for therapeutic benefit.
Reviewer 3 Report
The current review emphasizes on “unfolded protein response in cancer and immunogenic cell death”. In this paper the authors discussed how irreparable ER homeostasis can induce immunogenic cell death (ICD). Further they discussed the key aspects of the functional dichotomy of UPR in cancer cells and how this signal can be harnessed for therapeutic benefit in the context of ICD. The paper is exceptionally well written and organized.
Some points should be addressed before publication.
Major comments – 1) For the sections a) The pro-survival function of the UPR and b) UPR as a mechanism of cell death – Please provide a schematic representation of the mechanism.
2) Make a tabular column for the compounds that can modulate UPR pathways and prevent diseases.
1) Minor comments – At few places there are grammar errors, incorrect tense uses.
Author Response
Thank you very much for the comments. We’ve added another figure (Figure 2) to illustrate the two facets of UPR in cell survival and death.
Regarding the chemical compounds that modulate the UPR, we have described potential issue related to the use of UPR targeted drugs in the section “therapeutic outlook and conclusions”. We submit that whereas the request of the Reviewer to extend the discussion to other diseases where the UPR has been shown to play a role, is interesting, it is also beyond the scope of this Review that is focused on the role of UPR in the context of cancer and ICD.
Reviewer 4 Report
The review by Rufo et al deals with the unfolded protein response, an adaptation mechanism that senses perturbations in the ER and maintains ER homeostasis under physiological and pathological conditions. Depending on the ER protein burden, the UPR signaling pathways can promote either cell survival or cell death. In this regard, the authors bring the UPR in cancer cells into the main focus of the review and provide a mechanistic overview of the UPR-driven immunogenic cancer cell death (ICD) that is triggered by anticancer therapies. As the UPR provides a molecular switch between a protective and a killing modality (the ‘Yin and Yang’), targeting the UPR in tumor cells by different inhibitors for therapeutic benefits is challenging. Beside these compounds, ICD-based therapies as alternative anticancer strategies are discussed in the conclusive chapter of the review. The manuscript is well written and comprehensive and is supported with high quality illustrations. I have only a few minor points to the manuscript:
- Figure 1: As the authors rather use the more generic term IRE1, I think for the sake of consistency it makes sense to label this sensor as "IRE1", not "IRE1alpha" in the figure as well
- Line 64: Do the authors mean “to alleviate ER protein burden” ?
- Line 70: Do the authors rather mean “genes involved in ERAD and redox homeostasis and coding various chaperons” ?
- Line 86: “increase affinity towards phosphorylated eIF2alpha” ?
- Line 90ff, 97ff and 181ff: Please add reference(s)
- Line 99: “in the lung and gastrointestinal tract” ?
- Line 191: The abbreviation TRAIL could be defined in the text
- Line 214: a typo, ERMCs
- Line 222: “the pro-survival/apoptotic switch” ?
- Line 459: The abbreviation for photodynamic therapy (PDT) is not introduced in the text
- Figure 3 and line 466f: Do the authors mean dsRNA or ssRNA binding to TLR3? Please check.
- Line 525: “inhibiting the kinase auto-transphosphorylation and dimerization” ?
- “-s” is sometimes omitted or wrongly added to words/verbs throughout the text (e.g. “restore” in line 38, “possess” in line 43, “clients” in line 52, “operates” in line 221 etc)
Author Response
Thank you very much for the detailed comments. We’ve revised our manuscript accordingly.